# Effect of Targeted vs. Standard Fortification of Breast Milk on Growth and Development of Preterm Infants (≤32 Weeks): Results from an Interrupted Randomized Controlled Trial

**DOI:** 10.3390/nu15030619

**Published:** 2023-01-25

**Authors:** Joanna Seliga-Siwecka, Justyna Fiałkowska, Anna Chmielewska

**Affiliations:** 1Department of Neonatology and Neonatal Intensive Care, Medical University of Warsaw, 02–091 Warszawa, Poland; 2Department of Clinical Sciences, Pediatrics, Umeå University, 901 87 Umeå, Sweden

**Keywords:** breastmilk, fortification, preterm infant, targeted modification, neonatal intensive care unit, macronutrients, supplementation

## Abstract

Human milk is recommended for very low birth weight infants. Their nutritional needs are high, and the fortification of human milk is a standard procedure to optimize growth. Targeted fortification accounts for the variability in human milk composition. It has been a promising alternative to standard fixed-dose fortification, potentially improving short-term growth. In this trial, preterm infants (≤32 weeks of gestation) were randomized to receive human milk after standard fortification (HMF, Nutricia) or tailored fortification with modular components of proteins (Bebilon Bialko, Nutricia), carbohydrates (Polycal, Nutricia), and lipids (Calogen, Nutricia). The intervention started when preterms reached 80 mL/kg/day enteral feeds. Of the target number of 220 newborns, 39 were randomized. The trial was interrupted due to serious intolerance in five cases. There was no significant difference in velocity of weight gain during the supplementation period (primary outcome) in the tailored vs. standard fortification group: 27.01 ± 10.19 g/d vs. 25.84 ± 13.45 g/d, *p* = 0.0776. Length and head circumference were not significantly different between the groups. We found the feasibility of targeted fortification to be limited in neonatal intensive care unit practice. The trial was registered at clinicaltrials.gov NCT:03775785.

## 1. Introduction

Preterm birth results in a high risk of mortality and is a cause of several morbidities, including extrauterine growth restriction [1,2,3]. Human milk (HM) is the optimal source of nutrition for premature infants [4]. Despite numerous proven benefits, the concentration of some nutrients in HM may be too low to meet the high nutritional needs of premature infants. To ensure optimal growth and development, human milk fortification (HMF) is recommended for all very low birth weight (VLBW, <1500 g birth weight) infants [4,5].

There are several modalities for fortifying HM. *Standard fortification* provides a fixed dose of a compound fortifier added to breastmilk based on the predefined composition of HM to achieve recommended nutritional values and is most widely used in the neonatal intensive care unit (NICU) [6,7]. However, measurements of protein, glucose, and lipids show interindividual and intraindividual variations; hence, other approaches have been suggested [8]. In an *adjustable fortification* strategy, blood urea nitrogen concentrations serve as a surrogate for the response to protein supplementation, and protein fortification is adjusted accordingly. *Tailored/targeted human milk fortification* is achieved by adding only nutritional components (protein, carbohydrates, and fat) to HM based on the results of repeated breast milk composition bedside analyses [5].

While standard breastmilk fortification is the most feasible strategy in a NICU, it does not account for the substantial variability in HM composition between mothers and between milk samples from the same individual [9]. Not accounting for this variability may lead to inappropriate nutritional intake in one-third of all preterm infants [10]. The effects of targeted (individualized) HMF, encompassing adjustable and targeted breastmilk fortification, have recently been systematically reviewed by Fabrizio et al. [11]. The Cochrane review and meta-analysis concluded that individualized HMF improves the short-term velocity of weight, length, and head circumference [11]. However, the studies included in these analyses were not uniform. Researchers have chosen multiple fortification regimes, ranging from adding one to adding all three nutrients [12,13,14]. Furthermore, differences in unit staffing and workload between healthcare systems and macronutrient products make it difficult to draw a clear conclusion and implement the results worldwide.

To our knowledge, this is the first randomized controlled trial to evaluate whether targeted HMF can optimize growth in infants born at a gestational age < 32 weeks using all three macronutrients in an Eastern European setting. The study protocol was approved by the local ethics committee and was published [2]. The trial was registered at clinicaltrials.gov (NCT03775785).

### 1.1. Objectives

#### Research Hypothesis 

Tailored fortification of enteral nutrition improves weight gain velocity in preterm infants born at ≤32 weeks of gestation.

### 1.2. Study Objectives 

#### 1.2.1. Primary Objective 

The primary objective was to determine whether tailored compared to the standard fortification of enteral nutrition improved weight gain velocity in preterm infants born at ≤32 weeks of gestation. 

#### 1.2.2. Secondary Objectives 

Key Secondary Objectives 

The key secondary objective was to determine the following anthropometric parameters in preterm infants born at ≤32 weeks of gestation at discharge and 4 months: 

feeding tolerance, velocity of weight gain, length and head growth

## 2. Materials and Methods

### 2.1. Trial Design

The trial was designed as a randomized observer- and patient-blinded controlled multicenter superiority trial, with two parallel groups with a 1:1 allocation ratio.

### 2.2. Participants 

#### 2.2.1. Study Setting

The study was initially planned as a multicenter trial; however, two out of three centers failed to randomize patients due to staff shortages. Finally, the study was completed at the Neonatal and Intensive Care Department of the Medical University of Warsaw.

The study site was a level III teaching hospital with approximately 2500–3000 (100 ≤ 32 weeks of gestation) deliveries per year. The local protocol was based on the standard fortification of own mothers’ milk (OMM) and donor human milk (DHM).

#### 2.2.2. Eligibility Criteria

All parents of infants born at less than 32 weeks of gestation and admitted to the NICU were approached by one of the research team members within the first week of life (as full enteral feeding is usually reached at a minimum of 7 days of life). Recruitment was conducted between June 2019 and June 2022. After obtaining written consent for participation in the trial, the patient’s medical record number was immediately registered on a secure web-based platform, and demographic data were recorded.

#### 2.2.3. Inclusion Criteria 

Patients eligible for the trial had to comply with the following criteria at randomization: 

Gestational age at birth ≤ 32 weeks Enteral feeding of at least 80 mL/kg/day 50% donor or maternal milk-based enteral feeding Parenteral/legal guardian consent 

#### 2.2.4. Exclusion Criteria 

Formula feeding Small for gestational age (birth weight < 3rd percentile) Presence of congenital abnormalities, which increase the risk of necrotizing enterocolitis such as hypoplastic left heart syndrome, transposition of the great arteries, omphalocele, gastroschisisNecrotizing enterocolitis (NEC) Withdrawal of feeding > 7 days Sepsis Death 

#### 2.2.5. Obtaining Informed Consent

All parents of infants born at less than 32 weeks of gestation and admitted to the study site were approached by one of the research team members within the first seven days of life. They provided oral and written information about the study. Parents were allowed to participate in an informed discussion with the attending physician and study personnel. The research team members obtained written consent from parents willing to allow their children to participate in the trial. Information consent forms and information sheets were provided in Polish for all parents. Given the limited diversity of our population, we did not recruit newborns of foreign parents unless they spoke Polish on a level that allowed a full understanding of the study.

Regarding additional consent provisions for collecting and using participant data and biological specimens, we did not plan to collect samples for ancillary studies. 

#### 2.2.6. Sample Size

The sample size required to compare two means in a two-sided equality test was estimated based on results from a prior double-blind, randomized clinical trial, investigating the effect of TG vs. SF of breast milk on the changes of anthropometric parameters and body composition in preterm children [15,16]. It was determined that a mean difference of weight gain of 1.9 g/kg/day between groups would be clinically important and feasible during the intervention. The following assumptions were made for the calculation: type I error (α) 5%, power 80%, equal sample sizes in both groups, the mean weight gain in the standard fortification group 19.3 g/kg/day, and the mean weight gain in the target fortification group 21.2 g/kg/day. To account for the higher uncertainty in measured weight gain due to differences between the studied and the quoted trial population, the standard deviation value taken from the prior trial was increased by 50% to 3.75.

The estimated minimum size of each group was 68. Accounting for a presumed 20% attrition rate due to potential dropouts, deviations from the protocol and loss to follow-up, the minimum sample size required was estimated at 156 infants or 78 infants per treatment arm.

### 2.3. Interventions 

#### 2.3.1. Explanation for the Choice of Comparators

Standard fortification (SF) assumes that HM has a protein level of 1.5g/dL. HM, however, is highly variable in nutrient content, both between mothers and between samples from the same mother [10,11]. A recent study suggested that not considering this variability leads to inadequate intake in approximately 25–40% of VLBW infants due to low protein and energy content [12]. Nonetheless, it is the most widely used strategy for HMF, and thus, its choice as a comparator is logical.

#### 2.3.2. Intervention Description

After reaching 80 mL/kg/day of enteral feeding, patients were randomized to receive SF (Bebilon HMF, Nutricia^®^) or targeted fortification (TF) (protein: Bebilon Suplement Bialka, Nutricia^®^; lipids: Calogen, Nutricia^®^; carbohydrates: Polycal Nutricia^®^) [17]. The macro-and micronutrient contents have been previously published [17]. Milk fortification was routinely performed twice a day (at 8 am and 8 pm) for the following 12-h nursing shift. For the study, TF was integrated into this schedule and performed by experienced research nurses (RNs) [17].

One of the researchers (JSS) performed milk analysis in the NICU research laboratory at Princess Anna Mazowiecka Hospital three times per week (Monday/Wednesday/Friday) at 10:00 am and after protocol amendment twice per week (Tuesday/Thursday) from batches collected from the two previous days. A 10 mL aliquot from each batch of native breast milk was used for macronutrient analysis (Miris ^®^ HMA) per the protocol [17]. The remaining batch was fortified using a routine fortifier. Macronutrient analysis determined the amount of extra fat, protein, or carbohydrate needed in the batch to obtain the final target fortified breast milk (FBM).

The mean of three measurements per batch (3 × 2–3 mL) was used to calculate the required amount of extra fat, protein, and carbohydrate for the following three days of fortification using a predefined Excel spreadsheet (Microsoft Inc., Redmond, Washington, USA). Milk analysis was performed for both treatment arms; however, only the intervention group received TF.

The desired macronutrient concentration in breast milk was 4.4 g/100 mL of fat, 3 g/100 mL of protein, and 8.8 g/ 100 mL of carbohydrate to meet the European Society for Paediatric Gastroenterology, Hepatology, and Nutrition (ESPGHAN) guidelines (6.6 g/kg/d of fat, 4.5 g/kg/d of protein, and 13.2 g/kg/d of carbohydrate) assuming an intake of 150 mL/kg/d. 

Target fortification was conducted in three steps: 

Determination of macronutrient concentration in OMM/HDM. SF: Human milk fortifier, HMF Nutricia^®^. TF: Adding fat, protein, or carbohydrates to achieve the target levels of macronutrients. 

In cases where the macronutrient component after SF exceeded the target value, only other deficient macronutrient components were adjusted.

The patients were fed every 3 h via a gastric tube by RNs. Starting at 33 weeks of postconceptional age (PCA), non-nutritive sucking stimulation was initiated by occupational therapists. At approximately 34 weeks of PCA, infants transitioned to bottle feeding. When breastfeeding was established, patients received TF as one or two bottled feeds.

As a safety assessment to ensure that an appropriate amount of fortifier was added, the osmolality of unfortified and FBM samples was measured using a 3320 Micro-Osmometer (Advance Instruments, Norwood, MA, USA). Bedside nurses were informed whether the osmolality of fortified milk was within the acceptable target range (400–600 mOsmol/kg) before milk was administered during the next 12-h shift. Osmolality lower or higher than the defined target range was considered a sample preparation error of fortification, and the single-nutrient fortification was omitted. TF prescription was completed before noon. The attending physician approved this prescription. Subsequently, individual additives were provided by the nutrition services. Bedside nurses prepared batches of FBM, including additives for target fortification, and divided them into single feeding portions to be administered to infants [17].

The intervention continued until 37 weeks of PCA or hospital discharge. The parents, attending physicians, and outcome assessors were blinded to the interventions.

#### 2.3.3. Criteria for Discontinuing or Modifying Allocated Interventions

Criteria for discontinuing allocated intervention included

SepsisNECWithdrawal of parental/guardian consentPoor feeding tolerance, defined as increasing abdominal distension >2 cm between inter-observer measurement or regurgitations after feeding >3 feeds per day

#### 2.3.4. Strategies to Improve Adherence to Interventions

The medical notes of the infants included in the study were visibly marked to promote adherence to the study protocol. A flowchart explaining the inclusion, exclusion, and discontinuation criteria is available for the patient’s medical notes.

#### 2.3.5. Relevant Concomitant Care Permitted or Prohibited during the Trial

The participants continued to receive standard neonatal care. Interventions aimed at improving weight gain, such as increased daily intake (>160–170 mL/kg/day) or increased dosing of vitamin D (>1000 IU/L), or prescription of milk formula, were forbidden.

### 2.4. Outcomes

#### 2.4.1. Primary Outcome 

Weight gain velocity was measured starting from the day infants regained their birth weight to 4 weeks and then weekly until discharge. Length and head circumference were measured weekly until the patient was discharged.

#### 2.4.2. Secondary Outcomes 

Growth (weight, length, and head circumference) was assessed at discharge and four months of corrected age.Feeding tolerance under the whole fortification period. Morbidity: Incidence of NEC, retinopathy of prematurity (ROP), bronchopulmonary dysplasia (BPD), intraventricular hemorrhage (IVH), periventricular leukomalacia (PVL), sepsis, and pneumonia. The definitions are as follows:

The outcomes are defined as
Feeding tolerance was defined as hemorrhagic residuals or vomiting of bile until pathological causes were ruled out (intestinal obstruction or ileus) [18]. Gastric residuals and abdominal girth were not routinely assessed. Isolated green or yellow residuals were considered unimportant.NEC: Stage II or III. Stage II requires clinical manifestations of a distended abdomen and radiological verification (intramural or portal gases). Stage III requires findings like in Stage II and more severe clinical symptoms (shock, need for a respirator). In surgically verified cases, radiological verification is not required [19].ROP: Stages I to V, diagnosed by an ophthalmologist according to international criteria [20].BPD: Need for oxygen, continuous positive airway pressure (CPAP,) or mechanical ventilation at 36+0 weeks of gestational age [21].IVH as defined by Volpe [18].PVL as defined by Volpe [18].Early- and late-onset sepsis was defined as positive blood or cerebral fluid culture at less and more than 72 h of age, respectively [22].

The schedule of enrolment, interventions (including any run-ins and washouts), assessments, and visits for participants are presented in Table 1.

### 2.5. Recruitment

We planned to continue until a minimum of 200 valid observations were collected from every arm. As part of the antenatal consultation, women with threatened preterm labor were scheduled for a short meeting with a member of the recruitment team. During this appointment, they were offered to participate in the trial. To increase participant enrolment, the medical staff carried out a second patient screen during admission to the NICU. The enrolment period was extended from 2019 to 2022 (with intermittent withdrawals secondary to equipment failure). Recruitment rates were monitored monthly. In return, women were offered additional breastfeeding support by a certified lactation consultant, as reported previously [17]. 

### 2.6. Assignment of Interventions: Allocation 

#### 2.6.1. Sequence Generation

The allocation sequence was computer generated. Block randomization was performed with stratification by the delivery mode. Patients were randomly assigned to standard or tailored enteral nutrition fortification groups in a 1:1 ratio. The block size was varied and concealed until the primary endpoint analysis. 

#### 2.6.2. Concealment Mechanism

A member of the recruitment team approached caregivers within the infant’s first 7 days of life. They explained the study and obtained written consent for participation in the trial. Subsequently, the patient’s medical record number was registered on a secure web-based platform, and demographic data were recorded. The platform assigned a study number, together with the allocated treatment. 

#### 2.6.3. Implementation

A member [a physician not involved in patient care] of the research team prescribed the allocated fortification in the patient’s drug chart. Milk fortification was routinely conducted twice a day (at 8 am and 8 pm) for each following 12-h nursing shift. For the study, TF was integrated into this schedule and performed by experienced RNs [14]. The intervention was performed by an RN blinded to the treatment allocation. Patient data, along with the allocation results, were sent to the statistical team. The randomization list remained with the statistical team for the entire study duration. 

### 2.7. Assignment of Interventions: Blinding 

#### 2.7.1. Who Was Blinded

Bedside nurses, treating physicians, and clinical psychologists were blinded to the treatment allocation. Milk fortification was performed on a milk bank by an experienced RN. The prepared milk portions were transported to the unit. The feeding portions from both treatment arms did not differ in color or structure. 

#### 2.7.2. Procedure for Unblinding if Needed

The unblinding procedures were previously published. If unblinding was necessary, the allocation was disclosed to the treating physician.

### 2.8. Plans for Assessment and Collection of Outcomes

#### Primary Outcome 

Weight gain velocity was measured starting from the day infants regained their birth weight to 4 weeks, then weekly until discharge using the Seca 336 Baby Scale^®^. Length and head circumference were measured weekly until discharge using a Seca 336 baby measuring rod^®^.

### 2.9. Plans to Promote Participant Retention and Complete Follow-Up

All randomized infants who prematurely discontinued the study intervention were considered off-study drug/on-study. They followed the same participant timetable as the infants who continued the study treatment.

Once an infant was enrolled or randomized, the study site made every reasonable effort to follow the infant for the entire study period.

The participants could withdraw from the study for any reason at any time. The investigator could withdraw the participants from the study to protect their safety.

### 2.10. Data Management

All data collection was completed electronically. Data integrity was enforced through various mechanisms. Referential data rules, valid values, range checks, and consistency checks against data already stored in the database (i.e., longitudinal checks) were supported. Modifications to the data written in the database were documented through either a data change or inquiry system. Data entered into the database were retrieved for viewing through data entry applications. The type of activity an individual user may undertake is regulated by the privileges associated with their user identification code and password.

### 2.11. Confidentiality

Complete patient and study information was stored on a secure, password-protected web-based platform. Only the researchers involved in the study were provided with a personalized login and password to access the study information. The statistical team did not have access to sensitive data, such as date of birth, address, or contact details. All records containing the patient details and relevant medical histories were stored separately in a locked file cabinet.

There were no plans for the collection, laboratory evaluation, and storage of biological specimens for genetic or molecular analysis in this trial/future use. We did not plan to perform any genetic or molecular analysis in this trial. 

### 2.12. Statistical Methods for Primary and Secondary Outcomes

Baseline characteristics are presented according to the treatment groups. Categorical variables were presented as the number of counts and proportion of the group. Continuous variables are described as mean and standard deviation or median with interquartile range. Continuous variables were tested against the normality of distribution using the Shapiro–Wilk test and verified with skewness and kurtosis. In the justified cases, a visual assessment was performed. The equality of variance between groups was tested using Levene’s test. For continuous variables distributed normally with homogenous variances, the Student’s *t*-test was used to verify the differences in means. For continuous variables that were not normally distributed, comparisons were performed using the Mann–Whitney U test. Comparisons of groups for categorical variables were performed using the Pearson Chi-square test or Fisher’s exact test, as appropriate. In addition, the risk ratio and mean or median difference (MD) are presented, along with 95% confidence intervals (CIs). Comparisons in time were performed using the paired *t*-test or Wilcoxon test. All statistical calculations assumed *alpha* = 0.05 and were performed with R statistical software, version R-4.1.2.

## 3. Results

### 3.1. Demographics

Between 2019 and 2022, 392 infants born below 32 weeks of gestation were admitted to the NICU and screened for eligibility. Initially, 344 infants were excluded from the study for the following reasons: declined consent (*n* = 200), paused recruitment (*n* = 100), and failure to meet the inclusion criteria (*n* = 44). Fifty-five infants were initially randomized; however, 16 did not receive the allocated intervention. Thirty-nine singleton (*n* = 25) and twin (*n* = 7) births at a median age of 29 (range, 26–31) weeks and a mean birth weight of 1306 (±454.3) g were randomly assigned to SF (*n* = 21) or TF (*n* = 18) (Figure 1). Siblings from multiple pregnancies were randomly assigned to different treatment groups.

The baseline characteristics did not differ between the groups (Table 1).

### 3.2. Milk Composition

The average nutritional content of the breast milk was similar throughout the trial. There were no significant differences in baseline levels of macronutrients by the end of the first week of supplementation between the two groups: protein (2.0 ± 0.34 vs. 1.91 ± 0.22 g/100 mL, *p* = 0.328), glucose (7.60 vs. 7.40 g/100 mL, *p* = 0.219), lipids (3.48 ± 1.40 vs. 3.65 ± 1.20, *p* = 0.417), calories (71.06 ± 12.38 vs. 71.21 ± 12.04 kcal/100 mL, *p* = 0.972) (Table 2). Within each group, we noted differences in milk composition over the study period: protein concentration decreased significantly over the first four weeks [MD, 95% CI: −0.69 (−0.88, 0.50, *p* < 0.001 vs. −0.49 (−0.71, 0.27), *p* = 0.001], and the glucose concentration decreased over time in the standard group [MD, 95% CI: 0.49 (0.21, 0.77), *p* = 0.004] (Table 2).

### 3.3. Nutritional Intake and Growth

Only one infant in each group received formula feeding. The rates of OMM and DHM feeding did not differ between the groups. The mean achieving 80 mL/kg/d of enteral feeding did not differ between the groups (SF 6.44 vs. TF 7.9, *p* = 0.21) (Table 1). All infants in both groups received 1 g of HMF Bebilon Nenatal Nutricia^®^ per 25 mL of HM with a maximum dose of 6.6 g/kg. Forty-four percent (8 out of 18 newborns) required any type of nutrient throughout the study period. In the first week of the study, one infant required lipid supplementation only. Twenty-seven percent (5 out of 18) of patients required all three supplements for the entire study (Figure 2.)

The average weight gain in g/d was higher in the tailored group compared to the standard group (27.01 ± 10.19 g/d vs. 25.84 ± 13.45 g/d, respectively); however, no significant difference was found (*p* = 0.776). No significant difference in weight gain in g/kg/d between the tailored and standard groups was found (15.76 ± 3.10 g/kg/d vs. 16.84 ± 10.04 g/kg/d, respectively, *p* = 0.683). Differences between groups at the level of statistical tendency were identified in the case of length and head circumference gain in cm/wk (*p* = 0.056 and *p* = 0.074, respectively) (Table 3).

There was significant total weight gain in the tailored and standard groups over the first 4 weeks of supplementation [MD = 632.11, 95% CI (359.94, 904.28), *p* = 0.001 and MD = 601.78, 95% CI (414.70, 788.86), *p* < 0.001, respectively]. The change in the weight z-score over the same period was not significant in either group (*p* = 0.947 and *p* = 0.723, respectively) (Table 3).

### 3.4. Secondary Outcomes

Secondary outcomes, such as IVH (stages 1–4), late-onset sepsis, BPD, ROP, and PVL, were similar between the targeted and standard fortification groups. Only one infant in the control group developed necrotizing enterocolitis and was not fed enterally for seven days (Table 4).

## 4. Discussion

### 4.1. Principal Findings

In this prematurely terminated randomized controlled trial, we did not find any statistically significant difference in the velocity of weight gain during the supplementation period in infants born before 32 weeks of gestation who received TF compared to those who received SF. Changes in length and head circumference did not differ between the groups.

We ceased recruitment due to five cases of intolerance to feeds in the TF group, such as significant abdominal distention (reported in two consecutive measurements), regurgitation, and posseting. These adverse events were reported to the local bioethical committee according to the study protocol, and the researchers decided to terminate the trial [17]. Recruitment was interrupted after 39 infants were randomized. Consequently, the power for detecting the difference in weight gain velocity between the groups (primary outcome) decreased from 1.9 g/kg/day to 3.2 g/kg/day. Additionally, TF was found to be labor-intensive and time-consuming for both mothers and medical personnel. This was the reason why mothers withdrew from the study. It is worth noting that most parents lived a long distance from the hospital; given the epidemiological time (COVID pandemic), these families were faced with the emotional burden of being away from their babies. Additional obligations, such as participating in a study, are an additional source of anxiety. We noted significant difficulties among the mothers in complying with the milk collection protocol. Another confounder in the study was that weekly fortification was not compensated for the dilutionary effect of omitting samples with high osmolarity (if osmolality exceeded the safety range after adding all necessary supplements, the fortification was omitted for the planned study period).

### 4.2. Comparison with Other Studies

Evidence confirming that TF using all three macronutrients improves growth in preterm infants is limited. To date, ten studies have been published on TF [12] [10,19,20,21,22,23,24,25]. Three of the six randomized controlled trials evaluated the use of all three macronutrients. A Cochrane review published in 2020 concluded that TF improves growth in the preterm population. However, it included seven studies, of which only two evaluated individualized target fortification as an intervention [11]. Pooled results from these trials (72 participants) showed that the mean weight gain velocity in the TF group was higher by 2.49 g/kg/day (0.44–4.54) compared to adjustable fortification. Since then, only one new study by Rochow et al. has been published, confirming previous findings. These studies were performed at a Canadian academic center involving a large multidisciplinary research team [13,26]. Furthermore, three members of the team performed the HM analysis, which we found impossible to replicate in busy clinical settings. Our findings were confirmed in an Australian study, where the authors concluded that TF was time-consuming and labor-intensive and did not lead to growth improvement [12]. In Europe, dieticians are not part of the clinical team, which significantly increases the workload of the rest of the staff when it comes to TF. Studies conducted in Asian and European settings evaluated the addition of only one macronutrient (in most cases, protein), and this was found to be clinically feasible and improved growth [14,24,27,28]. It is also worth noting that, in contrast to Rochow et al.’s study, the HMF used in our study did not contain lipids; thus, in cases of low-fat HM concentrations, higher amounts of supplementation were required. This probably led to higher osmolality results compared to the Canadian study and might be the reason for the observed low tolerance to tailored supplementation [25].

In the study mentioned above by Rochow et al., weight gain velocity during the first 21 days of intervention was higher in the TF group compared to the SF group (MD 1.9 g/kg/day, CI 0.9, 2.9) [10]. The fact that we could not show the difference in the weight gain velocity is probably due to a lack of power. With the 39 included patients, we only had the power to detect differences as large as 3.2 g/kg/day. Regarding feasibility, the difference may be accounted for by economic differences between the countries where the study sites were located. In Canada, the GDP is more than twice as high as than in Poland [29].

### 4.3. Strengths and Limitations

A randomized controlled trial design is the methodology of choice for studying the effects of an intervention. Moreover, blinding of the parents, NICU staff, and outcome accessors minimized bias related to allocation, intervention, and outcome assessment. Transparency is an additional strength of our study; we registered the study on a public research platform and published the complete protocol in a peer-reviewed journal [17].

Evidence on the effect of TF on outcomes other than weight gain velocity is lacking; thus, we aimed to study whether neurodevelopmental scores will improve in the TF group at 12 and 24 months of corrected age. However, due to the early cessation of the study, we could not show the difference between the groups as calculated during planning.

The most important limitation of the current study was that it ceased before recruiting the target number of participants. Another limitation is that we did not measure body composition according to the protocol (funds were not obtained). Consequently, despite the effort and funds invested in designing and commencing the trial, we could not obtain results that would add to the existing evidence in the field.

However, it is important to emphasize that we identified several potential obstacles to introducing individualized fortification in clinical practice in the NICU. First, frequent measurements of milk composition (twice per week) significantly increased the workload. The potential solution might be to perform the measurement less often, for example, once a week instead of twice a week. Rochow et al. showed that measurements twice weekly led to a mean macronutrient intake within a range of ±5% of the targeted levels [13].

### 4.4. Further Research

An ongoing randomized controlled trial designed by Belfort et al. will study the effect of individualized fortification on growth, body composition, and development [30]. There are important differences between this study, Rochow et al.’s 2021 study, and the current trial [4,25]. The intervention will begin with achieving an enteral intake of 140 mL/kg, compared to 80 mL/kg. Another difference is that protein and fat, but not carbohydrates, will be added to HM in the experimental arm. Surprisingly, the target protein concentration was set at 1 g/100 mL (compared with 3 g/mL in our trial, which aligns with the ESPGHAN guidelines) [4]. Milk analysis will be performed daily. It is of value for research purposes, but based on available evidence, it is not necessary to achieve desirable macronutrient intakes [31]. The sample size (N = 130) will allow the authors to detect a moderate effect of the intervention on growth. Still, it is probably too small to detect subtle differences in developmental scores.

To date, only single-center studies have been conducted. Future research should focus on generalizability and various clinical scenarios, as feeding protocols differ between units and fortifiers differ between brands. The potential role of tailored enteral nutrition should be confirmed in multicenter international trials.

## 5. Conclusions

Targeted milk modification is a strategy that may allow for the optimization of growth in premature infants; however, feasibility and poor tolerance of feeds may be important obstacles in introducing this strategy to NICU clinical practice. Further research should focus on outcomes, such as body composition and development, and emphasize practical aspects.

## Figures and Tables

**Figure 1 nutrients-15-00619-f001:**
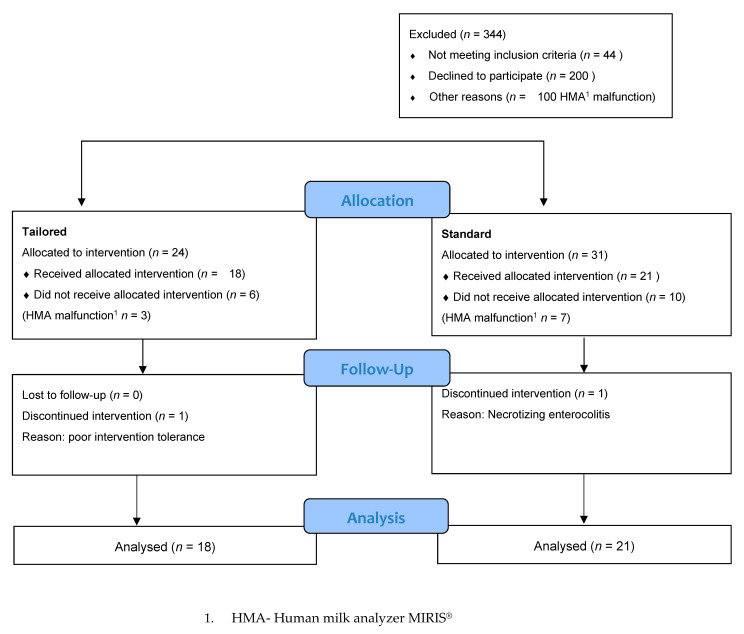
Participant enrollment flowchart.

**Figure 2 nutrients-15-00619-f002:**
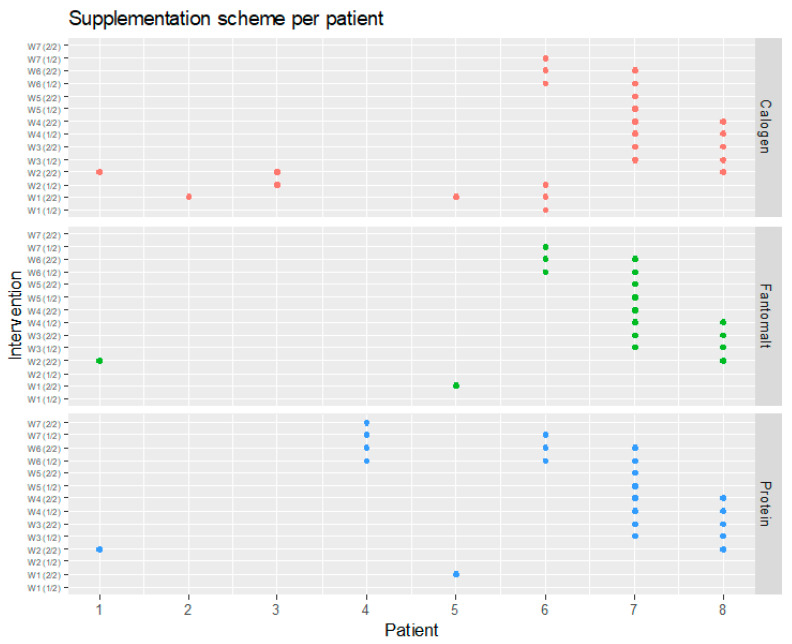
Supplementation per patient.

**Table 1 nutrients-15-00619-t001:** Baseline characteristics of study groups.

Variable	Tailored (*n* = 18)			Standard (*n* = 21)		
	*n* (%)	Mean ± SD	Median (Q1; Q3)	*n* (%)	Mean ± SD	Median (Q1; Q3)
Gender, male	9 (50.0)	-	-	10 (47.6)	-	-
Gestational week	-	29.22 ± 2.02	29.50 (28.00; 31.00)	-	28.24 ± 2.62	29.00 (26.00; 31.00)
Birth weight, g	-	1361.11 ± 454.30	1315.00 (1047.50; 1685.00)	-	1250.95 ± 429.68	1200.00 (1000.00; 1620.00)
Birth length, cm	-	41.33 ± 5.69	43.00 (37.50; 45.00)	-	38.88 ± 6.14	39.00 (35.00; 45.00)
Head circumference, cm	-	25.89 ± 7.21	28.25 (25.00; 30.00)	-	25.40 ± 6.58	27.00 (24.00; 29.00)
Birth weight z-score	-	0.39 ± 1.17	0.58 (-0.74; 1.13)	-	0.62 ± 1.19	0.54 (−0.06; 1.21)
Birth length z-score	-	1.42 ± 1.63	1.79 (0.22; 2.67)	-	1.03 ± 1.72	1.21 (−0.03; 2.14)
Head circumference z-score	-	0.55 ± 1.39	0.68 (-0.34; 1.64)	-	0.68 ± 1.09	0.98 (−0.05; 1.70)
80 mL/kg/day of maternal or human donor milk, day	-	6.44 ± 4.67	6.00 (4.00; 6.75)	-	7.90 ± 5.91	6.00 (5.00; 9.00)
50% donor or maternal milk-based enteral feeding	18 (100.0)	-	-	21 (100.0)	-	-
Milk type/source–mother	17 (94.4)	-	-	21 (100.0)	-	-
Milk type/source–formula	1 (5.6)	-	-	1 (4.8)	-	-
Milk type/source–donor	13 (72.2)	-	-	13 (61.9)	-	-

**Table 2 nutrients-15-00619-t002:** Milk composition during supplementation period.

Variable	Tailored	Standard	MD (95% CI)	*p*
Milk volume, mL				
Day 0	-	-	-	-
Week 1 (1/2)	241.60 ± 78.80	220.39 ± 65.79	21.21 (−30.10; 72.52)	0.406
Week 4 (2/2)	305.60 ± 106.79	264.00 ± 114.45	41.60 (−62.39; 145.59)	0.412
Last measurement	323.29 ± 114.93	313.60 ± 127.25	9.69 (−71.86; 91.25)	0.811
No. of HMF pieces				
Day 0	-	-	-	-
Week 1 (1/2)	8.94 ± 9.33	8.55 ± 7.94	0.38 (−5.55; 6.32)	0.896
Week 4 (2/2)	20.00 ± 7.48	17.20 ± 8.70	2.80 (−4.83; 10.43)	0.450
Last measurement	21.21 ± 7.43	16.50 ± 8.45	4.71 (−0.65; 10.06)	0.083
Protein, g/100 mL				
Day 0	2.02 ± 0.34	-	-	-
Week 1 (1/2)	2.00 ± 0.34	1.91 ± 0.22	0.09 (−0.10; 0.29)	0.328
Week 4 (2/2)	1.37 ± 0.24	1.32 ± 0.15	0.05 (−0.14; 0.24)	0.582
Last measurement	1.34 ± 0.26	1.38 ± 0.25	−0.04 (−0.21; 0.13)	0.640
Glucose, g/100 mL				
Day 0	7.50 (7.10; 7.80)	-	-	-
Week 1 (1/2)	7.60 (7.07; 7.80)	7.40 (7.05; 7.60)	0.20 (−0.10; 0.50)	0.219
Week 4 (2/2)	7.60 (7.53; 7.77)	7.90 (7.67; 8.00)	−0.30 (−0.50; 0.00)	**0.039**
Week 4 (2/2) (w/o one patient)	7.60 (7.60; 7.80)	7.90 (7.67; 8.00)	−0.30 (−0.40; 0.00)	0.069
Week 4 (2/2) (w/o one patient)	7.67 ± 0.18	7.86 ± 0.22	−0.19 (−0.39; 0.00)	0.051
Last measurement	7.70 (7.60; 7.80)	7.80 (7.60; 7.90)	−0.10 (-0.20; 0.10)	0.348
Fat, g/100 mL				
Day 0	3.36 ± 1.44	-	-	-
Week 1 (1/2)	3.48 ± 1.40	3.65 ± 1.20	8.00 (−17.00; 8.00)	0.417
Week 4 (2/2)	4.06 ± 1.56	3.82 ± 0.56	−0.10 (−0.80; 0.80)	0.940
Last measurement	4.19 ± 1.48	3.52 ± 1.05	0.68 (−0.17; 1.53)	0.113
Energy, kcal/100 mL				
Day 0	-	-	-	-
Week 1 (1/2)	71.06 ± 12.38	71.21 ± 12.04	−0.15 (-8.57; 8.27)	0.972
Week 4 (2/2)	73.50 ± 12.20	73.10 ± 4.93	0.40 (−8.35; 9.15)	0.925
Last measurement	75.35 ± 12.18	69.20 ± 10.06	6.15 (−1.27; 13.57)	0.101

HMF -human milk fortifier.

**Table 3 nutrients-15-00619-t003:** Weight, length, and head circumference development for the first 4 weeks.

Variable	Week 1/at Birth	Week 4/Study End	MD (95% CI)	*p*
Week 1 to week 4, tailored group (*n* = 8)				
Weight, g	1379.56 ± 441.49	2011.67 ± 647.56	632.11 (359.94; 904.28)	**0.001**
Weight z-score	0.00 ± 1.14	0.02 ± 0.89	0.02 (−0.72; 0.76)	0.947
Week 1 to week 4, standard group (*n* = 8)				
Weight, g	1264.56 ± 365.98	1866.33 ± 570.42	601.78 (414.70; 788.86)	**<0.001**
Weight z-score	0.24 ± 1.15	0.29 ± 1.06	0.05 (−0.26; 0.36)	0.723
At birth to study end, tailored group (*n* = 18)				
Weight, g	1361.11 ± 454.30	2253.22 ± 838.63	892.11 (548.76; 1235.46)	**<0.001**
Length, cm	41.33 ± 5.69	48.50 ± 6.02	7.17 (4.81; 9.53)	**<0.001**
Head circumference, cm	25.89 ± 7.21	32.53 ± 3.06	6.64 (3.34; 9.93)	**0.001**
Weight z-score	0.39 ± 1.17	−0.38 ± 1.35	−0.78 (−1.05; −0.50)	**<0.001**
Length z-score	1.42 ± 1.63	1.30 ± 1.91	−0.12 (−0.88; 0.64)	0.749
Head circumference z-score	0.55 ± 1.39	0.74 ± 1.51	0.19 (−0.55; 0.93)	0.592
At birth to study end, standard group (*n* = 21)				
Weight, g	1250.95 ± 429.68	2041.48 ± 843.51	790.52 (465.30; 1115.75)	**<0.001**
Length, cm	39.42 ± 5.76	44.90 ± 7.10	5.48 (3.37; 7.58)	**<0.001**
Head circumference, cm	25.42 ± 6.75	30.70 ± 4.05	5.28 (1.71; 8.84)	**0.006**
Weight z-score	0.62 ± 1.19	−0.35 ± 1.21	−0.97 (−1.32; −0.61)	**<0.001**
Length z-score	1.21 ± 1.55	0.56 ± 1.80	−0.65 (−1.13; −0.18)	**0.009**
Head circumference z-score	0.66 ± 1.12	−0.05 ± 1.43	−0.72 (−1.15; −0.28)	**0.003**

Data presented as mean ± standard deviation. MD–mean (week 4/study end vs. week 1/at birth), CI–confidence interval. Measurements compared with Student’s *t*-test for dependent groups.

**Table 4 nutrients-15-00619-t004:** Secondary outcomes.

Variable	Tailored (*n* = 18)	Standard (*n* = 21)	*p*
Poor feeding tolerance	6 (33)	3 (14.3)	
Sepsis	0 (0.0)	0 (0.0)	-
Necrotizing enterocolitis	0 (0.0)	1 (4.8)	>0.999
Intraventricular haemorrhage 1	0 (0.0)	1 (4.8)	>0.999
Intraventricular haemorrhage 2	2 (11.1)	6 (28.6)	0.247
Intraventricular haemorrhage 3	0 (0.0)	2 (9.5)	0.490
Periventricular leukomalacia	0 (0.0)	0 (0.0)	-
Bronchopulmonary dysplasia	3 (17.7)	8 (38.1)	0.260
Retinopathy of premature	7 (38.9)	7 (33.3)	0.980
Death	0 (0.0)	0 (0.0)	-
Late onset sepsis	0 (0.0)	1 (4.8)	>0.999
Enteral feeding suspended for at least 7 days	1 (5.6)	1 (4.8)	>0.999
>50% formula feeding	2 (11.1)	1 (4.8)	0.586

Data presented as n (% of group). Groups compared with Fisher’s exact test or Pearson’s Chi-square test ^1^, as appropriate.

## Data Availability

Data are available upon reasonable request from the corresponding author.

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
