# Peer review of "Effect of Targeted vs. Standard Fortification of Breast Milk on Growth and Development of Preterm Infants (≤32 Weeks): Results from an Interrupted Randomized Controlled Trial"

_nutrients, 2023, doi:10.3390/nu15030619_

Round 1

Reviewer 1 Report

The authors conducted a single center randomized observer and patient-blinded controlled trial comparing the effect of targeted vs standard fortification of breast milk in preterm infants born at ≤ 32 weeks of gestation. The study is significantly limited by a small sample size and interrupted recruitment. 

Major comments:

1. Line 368: 100 patients were excluded due to HMA malfunction and only 55 patients ended up being included in the randomization. It appears to me that the human milk analyzer only worked approximately one third of the time (55/155). Can the authors please comment on this? Additionally, even after randomization, 10 additional patients were excluded due to HMA malfunction (3 from tailored and 7 from standard group), which is a very significant number of patients given that eventually only 39 patients were included in the analysis. 

2. The authors need to clearly state in the methods section how they calculated the sample size needed for their RCT. The final sample size is small and makes the study underpowered to detect differences between groups.

Author Response

Dear Reviewer,

Thank you for your review of our paper and for your helpful comments.

Line 368: 100 patients were excluded due to HMA malfunction and only 55 patients ended up being included in the randomization. It appears to me that the human milk analyzer only worked approximately one third of the time (55/155). Can the authors please comment on this? Additionally, even after randomization, 10 additional patients were excluded due to HMA malfunction (3 from tailored and 7 from standard group), which is a very significant number of patients given that eventually only 39 patients were included in the analysis. 

Unfortunately, HMA malfunction developed while 10 patients were recruited into the trial and were receiving allocated treatment. Following which, we were forced to cease the study and ship the analyzer back to the manufacturer for repair. During this closedown period 100 potential participants who met inclusion criteria could not be recruited due to lack of equipment.

The authors need to clearly state in the methods section how they calculated the sample size needed for their RCT. The final sample size is small and makes the study underpowered to detect differences between groups.

We thank the Reviewer for this comment.

As stated in the manuscript, we previously published the study protocol, hence, we decided to limit the methods section to new information only (Seliga-Siwecka J, Chmielewska A, Jasińska K. Effect of targeted vs standard fortification of breast milk on growth and development of preterm infants (≤ 32 weeks): study protocol for a randomized controlled trial. Trials. 2020 Nov 23;21(1):946. doi: 10.1186/s13063-020-04841-x. PMID: 33225961; PMCID: PMC7682103.). However, we appreciate that this may cause confusion. Following the Reviewers’ comment, we chose to include more information on how the sample was calculated. The reviewed section has been highlighted in yellow.

Reviewer 2 Report

Thank you for your work. Human milk fortification is a significant aspect of care of preterm neonates. Targeted fortification has gained interest as a potential intervention aiming to improve growth. Despite early interruption, this randomized clinical trial provides useful information concerning fortification process. The manuscript is well-written, organized, and concise, and the study design was robust. Further research is certainly warranted, but this manuscript raises some important questions and underlines possible explanations for failure of trial completion. Crucial issues regarding individualized fortification in everyday clinical practice in NICUs are addressed.

Author Response

Dear Reviewer,

Thank you for taking the time to review our manuscript and your thorough comments.

We have made the suggested changes, which have been highlighted in green.

Round 2

Reviewer 1 Report

For future reference, the authors should note that the CONSORT guidelines (https://www.consort-statement.org/) should be followed for reporting any randomized controlled trials and one of the elements in the guidelines is that the authors should report how sample size was determined. 

The authors have adequately addressed my comments and revised their manuscript.